# Chronotherapeutics for Solid Tumors

**DOI:** 10.3390/pharmaceutics15082023

**Published:** 2023-07-26

**Authors:** Claire O. Kisamore, Brittany D. Elliott, A. Courtney DeVries, Randy J. Nelson, William H. Walker

**Affiliations:** 1Department of Neuroscience, Rockefeller Neuroscience Institute, West Virginia University, Morgantown, WV 26506, USA; cok0002@mix.wvu.edu (C.O.K.); bdh00006@mix.wvu.edu (B.D.E.); courtney.devries@hsc.wvu.edu (A.C.D.); randy.nelson@hsc.wvu.edu (R.J.N.); 2Department of Medicine, West Virginia University, Morgantown, WV 26506, USA; 3West Virginia University Cancer Institute, Morgantown, WV 26506, USA

**Keywords:** circadian rhythms, cancer, chronotherapy, chemotherapy, radiotherapy, immunotherapy

## Abstract

Circadian rhythms are internal manifestations of the 24-h solar day that allow for synchronization of biological and behavioral processes to the external solar day. This precise regulation of physiology and behavior improves adaptive function and survival. Chronotherapy takes advantage of circadian rhythms in physiological processes to optimize the timing of drug administration to achieve maximal therapeutic efficacy and minimize negative side effects. Chronotherapy for cancer treatment was first demonstrated to be beneficial more than five decades ago and has favorable effects across diverse cancer types. However, implementation of chronotherapy in clinic remains limited. The present review examines the evidence for chronotherapeutic treatment for solid tumors. Specifically, studies examining chrono-chemotherapy, chrono-radiotherapy, and alternative chronotherapeutics (e.g., hormone therapy, TKIs, antiangiogenic therapy, immunotherapy) are discussed. In addition, we propose areas of needed research and identify challenges in the field that remain to be addressed.

## 1. Introduction

Circadian (Latin; *circa* meaning around, *diem* meaning day) rhythms are the endogenous timekeeping mechanisms of living organisms. These approximately 24 h cycles drive important interactions with the environment (e.g., sleep–wake cycle, digestion, immunity, etc.) and are entrained via environmental cues (i.e., zeitgebers) such as light, temperature, food intake, and physical activity [1,2]. In mammals, blue light (~480 nm) stimulates the photopigment melanopsin within intrinsically photosensitive retinal ganglion cells (ipRGCs) during the day [3]. Melanopsin is a G protein-coupled receptor (GPCR) that, when stimulated, causes membrane depolarization leading to the release of the neuroexcitatory transmitter glutamate, which in turn activates neurons within the suprachiasmatic nucleus (SCN) [4,5]. The SCN, termed the central pacemaker of biological rhythms, consists of two bilateral nuclei located within the anterior region of the hypothalamus directly above the optic chiasm (Figure 1). When cells within the SCN are stimulated via the ipRGC relay, Ca^2+^ signaling is increased and ultimately results in expression of period genes, which form a crucial part of the transcription–translation feedback loops that drive the molecular clock [6,7,8,9].

The molecular transcription–translation feedback loops within the SCN are the main drivers of intrinsic rhythmic timekeeping. They involve the heterodimeric complex of the cytoplasmic proteins known as circadian locomotor output cycle kaput (CLOCK) and brain and muscle ARNT-like protein 1 (BMAL1). In the primary circadian transcription-translation loop, this complex translocates to the nucleus where it binds enhancer boxes (E-boxes) that drive expression of period 1, 2, and 3 (*PER1*, *PER2*, *PER3*) and cryptochrome 1 and 2 (*CRY1*, *CRY2*) [7]. PER and CRY proteins heterodimerize and translocate to the nucleus, where they inhibit CLOCK/BMAL1 regulation of their own expression. As cellular concentrations of PER and CRY proteins start to decrease due to ubiquitination, negative regulation of CLOCK/BMAL1 also decreases and the cycle, lasting approximately 24 h, starts over (Figure 2). The CLOCK/BMAL1 complex can also drive the expression of the nuclear receptors REV-ERBα/β, which repress BMAL1 expression, and RORα/γ, which drive BMAL1 expression (Figure 2) [7,10]. The oscillations in clock transcription factors allow for expression of target genes in cycles in which peak (zenith) and trough (nadir) expression are ~24 h apart. However, the oscillations of each target gene are unique and can vary depending on tissue type. Consequently, mutations in key clock genes can reduce amplitude (difference between the zenith and nadir), alter period length (i.e., one full cycle), or completely ablate oscillations in target gene expression [8,11,12]. Although almost all mammalian cells have core clock machinery, only the neurons of the SCN receive direct environmental cues (i.e., light), making the SCN the central regulator that ultimately controls circadian oscillations in all peripheral tissue (Figure 2) [12]. Without the hierarchical regulation by the SCN, individuals would have differing free-running periods ranging from 22–26 h.

Components of the molecular clock are therapeutic targets for cancer treatment due to their substantial involvement in the regulation of the cell cycle. For example, BMAL1 and PER2 influence daily oscillations of p53, a crucial tumor-suppressing protein. Deletion or mutation of one or both of these core clock genes therefore alters the cell cycle, which is partially regulated by p53. Additionally, p53 can regulate PER2 expression, which further implicates this dysregulated feedback loop in tumor growth. Similar disruptions in DNA replication can also promote tumorigenesis, as seen in murine models in which CLOCK-deficient mice have truncated telomeres [13,14]. Furthermore, c-Myc, a proto-oncogene and key regulatory transcription factor in the cell cycle, is regulated by the CLOCK/BMAL1 complex as well as other CLOCK-associated heterodimeric transcription factors [15,16]. PER-associated heterodimeric transcription factors also play a crucial role in the cell cycle, as evidenced by their oscillatory regulation of p16, a tumor suppressor protein [17]. The nuclear receptors ROR/REV-ERB contribute to cell cycle homeostasis through regulation of p21, which helps to guide the G1 and S phases, specifically, via downstream signaling [16,18]. Moreover, circadian dysregulation of DNA repair can also have pro-oncogenic effects. For example, DNA damage response mechanisms involving the cell cycle kinases ataxia telangiectasia mutated (ATM) and checkpoint kinase 2 (CHK2) are known to contribute to the development of colorectal cancer. Specifically, the over-expression of PER1, which interacts with ATM and CHK2 (both of which regulate cell cycle checkpoints), induces damage-associated apoptosis. Conversely, diminished PER1 expression (sometimes observed in human cancer tissue) can lead to survival of DNA-damaged cells that would otherwise undergo apoptosis [19,20]. Drugs targeting ROR, REV-ERB, and CRY proteins are currently in the early stages of research. However, more data and clinical trials are needed to integrate therapeutic targets for clock genes in tandem with traditional cancer therapies [14].

Perturbations of the circadian cycle alter many biological processes including metabolism, endocrine regulation, reproduction, cardiovascular function, sleep, and immunity [7,21,22]. Because of the tight regulation of circadian oscillations in many crucial proteins, chronic disruption of an individual’s internal clock leads to serious health issues, including cancer. The International Agency for Research on Cancer and the National Toxicology Program (United States) have officially classified night shift work as a likely carcinogen [23]. Breast and prostate cancer are among the most likely outcomes of proto-oncogenic circadian disruption due to the estrogen and aromatase-suppressing properties of melatonin [23,24]. Indeed, circadian rhythm disruption is a risk factor for breast cancer development. For example, circadian rhythm disruption via artificial light at night (ALAN) or chronic jet lag promotes mammary gland associated tumor growth and development in rodents [25,26,27,28]. Furthermore, epidemiological evidence suggests that there is a positive association between ALAN exposure and breast cancer in humans [16].

Whereas the detriments of circadian rhythm disruption on physiological and psychological health have been well established, circadian manipulation can be used therapeutically to optimize treatments for diseases such as cancer. Chronotherapy, the concept of optimizing the timing of drug administration to maximize therapeutic efficacy and minimize side effects, can be implemented for the treatment of solid tumors. Chronotherapeutics were first explored in cancer research in the early 1970s. In a pioneering chrono-chemotherapeutic study, time of survival was significantly increased in a murine model of leukemia by adjusting the treatment regimen to take advantage of daily oscillations in the maximal tolerated dose of chemotherapeutics [29]. In this study, all mice were given a total of 240 mg/kg of arabinosyl cytosine (ara-C) over a 24 h period. The treatment group based on endogenous circadian oscillations was given the highest doses earlier on in the cycle when the mice were the most tolerant (i.e., the first four doses were over two times higher than the last four doses in the 24 h cycle) instead of the standard 30 mg/kg every three hours. Not only did mice in the chronotherapeutic treatment group survive significantly longer, but they were tolerant of higher chemotherapy doses; in other words, exploiting time of administration produced more efficacious treatment [29]. Implementation of chronotherapy in cancer treatment likely has been hampered by greater success in ameliorating negative side effects of treatments than increasing their efficacy [30]. However, even improving the tolerability of treatment is crucial for promoting adherence to the treatment plan and enhancing quality of life for cancer survivors [31]. More research is needed to further elucidate how chronotherapeutic strategies can be utilized in tandem with traditional cancer treatments to achieve a desired synergistic effect in achieving remission. Below, we review progress in the development of chronotherapeutic strategies for the treatment of solid tumors as well as discuss further research questions presented by gaps in the field.

## 2. Chrono-Pharmacology in Solid Tumor Treatment

The study of chronotherapy is inherently interdisciplinary, as chronobiological principles are applied to several different fields such as cancer, immunology, infectious diseases, and more. Arguably, some of the most crucial variables to consider when applying a chronobiological approach to the treatment of any disease fall within the realm of pharmacology. Chrono-pharmacology is the study of how daily rhythms in drug availability are driven by diurnal variation in drug metabolism, absorption, kinetics, and other characteristics. Because many key proteins with functions in metabolic processes such as oxidation/reduction reactions, hydrolysis, conjugation, acetylation, xenobiotic clearance, and many more are expressed as daily oscillations, optimal timing of drug delivery significantly reduces toxicity of common anti-cancer drugs (e.g., docetaxel and cisplatin) in rodents [32,33,34]. Metabolic proteins, as well as molecular anti-cancer drug targets controlled by clock genes, have been extensively reviewed [32].

Daily oscillations of metabolic enzymes in peripheral tissue can be regulated locally, resulting in rhythms that are unique to a specific organ. Processing of xenobiotic substances (i.e., drugs) typically occurs in the liver, kidneys, and gastrointestinal tract. Metabolic reactions in these areas determine key processes such as methods and rates of absorption and secretion that are unique to each drug. Steps of drug metabolism are categorized into Phase I (oxidation, reduction, and hydrolysis), Phase II (conjugation), and Phase III (excretion). Enzymes involved in each phase are regulated by the proline–acidic amino acid-rich basic leucine zipper (PARbZip) family of transcription factors including albumin D-site-binding protein (DBP), hepatic leukemia factor (HLF), and thyrotroph embryonic factor (TEF). DBP, HLF, and TEF are regulated by molecular clocks at the organ level in the liver, kidneys, and small intestine, indicating that drug metabolism and therefore toxicity inherently fluctuate throughout the day [32,35]. Indeed, one study reported that mice deficient in all three of these transcription factors (DBP, HLF, and TEF triple knockout) took significantly longer to clear 25 mg/kg of pentobarbital than WT mice at both zeitgeber time (ZT) 4 and ZT16. ZT is a unit of time based on the period of an entrainment signal (i.e., zeitgeber). ZT0 refers to the onset of activity that is typically coupled to the beginning of the light period in diurnal species. Knockout mice also exhibited disrupted phases (timing of rhythms compared to a reference) and aberrant amplitudes, but not reductions in target gene mRNA expression, further suggesting that circadian control indirectly affects the timing of metabolic functions. Furthermore, the study demonstrated an increased survival in WT-naïve mice at two weeks post-treatment with both mitoxantrone and cyclophosphamide (chemotherapeutics) when compared to triple knockout mice that received the same treatments [36].

Nuclear receptors, which act as transcription factors when activated by lipid-soluble ligands (e.g., estrogen receptors), have the ability to drive expression of phase I, II, and III metabolic enzymes upon binding xenobiotic substances. As previously described, the nuclear receptors REV-ERBα/β and RORα/γ are directly involved in feedback loops that control BMAL1 expression. The expression of these and many other nuclear receptors is also controlled by the molecular clock. Data from mice indicate that cryptochromes (CRY1 and CRY2) physically interact with approximately one-third of all murine nuclear receptors invariably, as well as variably with an additional third of nuclear receptors [37]. Therefore, CRY1 and CRY2 have the potential to modulate about two-thirds of murine nuclear receptors. Furthermore, the same study reports that among the strongest interactions with CRY proteins are the xenobiotic-detecting nuclear receptors pregnane X receptor (PXR) and constitutive androstane receptor (CAR), which are repressed by CRY1 and CRY2 [37]. These data indicate an indirect consequence of the molecular clock in the metabolism of xenobiotic substances. Based on this information and what is known about the transcription–translation feedback loop of the molecular clock, it is plausible to speculate that the metabolism of chemotherapeutics would be more rapid and efficient when CLOCK and BMAL1 are highly expressed and CRY1 and CRY2 are not. Indeed, *bmal1^−/−^* mice were more sensitive to the toxicity of cyclophosphamide [37]. Conversely, cryptochrome double knockout mice (*cry1^−/−^, cry2^−/−^*) displayed a decreased sensitivity to the toxic effects of the drug when compared to their WT littermates. These data reveal a deviance from the typical circadian regulation of drug toxicity in WT mice [38].

CAR and peroxisome proliferator-activated receptor alpha (PPAR-α), a nuclear receptor involved in lipid metabolism, partially control the daily oscillations in expression of cytochrome P450 (CYP450) enzymes and ATP-binding cassette (ABC) transporters. CYP450 enzymes and ABC transporters (efflux pumps) both play crucial roles in xenobiotic metabolism and toxicity [32,39]. Therefore, their rhythmic expression is greatly considered when designing anti-cancer chronotherapeutic regimens. For example, the CYP450 enzyme Cyp3a has a role in the circadian regulation of tolerance to some anti-cancer drugs that undergo oxidation during their metabolism [40]. Furthermore, Pulido et al. demonstrated that circadian oscillations in P-glycoprotein (Pgp), a common ABC transporter, play a role in daily rhythms seen in blood–brain barrier permeability [11]. Although we discuss only a few instances of circadian control of metabolic processes, there exist many more examples, yet we still lack the translational skills to apply these principles broadly to the treatment of solid tumors in humans. Given the complexities of peripheral oscillations in metabolic enzymes, the pharmacokinetics of individual anti-cancer drugs should be assessed when considering chronotherapy. For example, gemcitabine and L-alanosine are two chemotherapeutics of the antimetabolite class, yet the optimal times of administration are almost twelve hours apart from each other in WT-naive B6D2F1 mice [40]. Although chronotherapeutic clinical trials have been conducted with common anti-cancer drugs, more translationally focused research is needed to bridge the gap between controlled laboratory experiments and the clinic.

## 3. Chrono-Chemotherapy for Solid Tumor Treatment

One of the most commonly utilized treatment modalities for cancer is chemotherapy. As of 2018, nearly 10 million individuals were on a regimen involving chemotherapy, a number which is expected to increase 53% by 2040 as the population ages [41]. First explored in research nearly 50 years ago, the goal of chrono-chemotherapy is to deliver medication at a point in the circadian cycle that maximizes the efficacy of the drug while reducing treatment-related adverse effects [42]. Although the concept of chrono-chemotherapy is decades old, only a relatively small number of studies have explored this promising therapeutic tool.

Chemotherapy works by exerting cytotoxic effects, thus killing tumor cells, and slowing cancer progression. Chemotherapeutics can be used as a primary anti-cancer treatment (neoadjuvant), simultaneously, or following other treatment modalities to prevent reoccurrence (adjuvant). Many chemotherapy drugs exist, capitalizing on various mechanisms of action. For example, alkylating agents, such as cyclophosphamide, cisplatin, carmustine, and thiotepa, contain the alkaline compound R-CH^2+^. The addition of this group targets tumor growth by suppressing the replication and transcription of DNA [43]. Other common classes of chemotherapeutic agents include anthracyclines (doxorubicin), antimetabolites (fluorouracil), and antibiotics (actinomycin) [43]. Although they will not be covered in depth here, the mechanisms by which each drug exerts their effects are extensively detailed [44]. In general, chemotherapy functions by preventing the biosynthesis of DNA, RNA, or nucleic protein, which alters the genetic landscape and in turn reduces tumorigenesis [44]. Unfortunately, this process also results in the genetic instability of healthy cells and an increased likelihood of future mutations [45]. However, it has been proposed that the mechanism of action of each of these drugs may vary based on the time of day in which the drug is administered [30]. Thus, administering chemotherapy in accordance with an optimal time of day may limit potentially harmful genomic effects. The toxicity of chemotherapeutic drugs is moderated by several factors pertaining to circadian control, such as drug metabolism, time-dependent target expression, and DNA repair [46]. For example, cisplatin facilitates the production of DNA-damaging agents that disrupt the nucleotide excision repair system. Because the rate-limiting DNA damage recognition factor, Xeroderma pigmentosum A (XPA), is expressed in daily oscillations, cisplatin is a good candidate for chrono-chemotherapy (i.e., time of treatment might align with the XPA batiphase) [47]. Given that the side effects of chemotherapy medication may also differ as a result of their mechanism and time of dose administration [44], it is perhaps not surprising that the use of chrono-chemotherapy may result in reduced adverse side effects experienced by the patient. Indeed, chronotherapeutics are a useful tool in minimizing the toxicity, and subsequent tolerability, of chemotherapy medication [48,49,50,51]. This is of particular clinical importance, as not only is the anti-tumor activity of a drug the greatest at the point when it is best tolerated, but up to 50% more anti-cancer medication can be administered at this time as well [52].

In an early study [53], 31 patients with advanced-stage ovarian cancer were administered two chemotherapy medications (doxorubicin and cisplatin) 12 h apart, beginning at either 0600 h or 1800 h. Patients assigned to morning doxorubicin followed by evening cisplatin administration experienced improved tolerance of the chemotherapeutics, as evidenced by a decreased need to lower their dosage and fewer treatment-related complications [53]. Because it has been reported that the anti-tumor efficacy of the drug is correlated with the dose of the drug administered [44], presumably, a more aggressive treatment regimen made permissible by increased tolerance of the patient will likely result in greater overall success. A study examining the timed administration of cisplatin in 41 non-small cell lung cancer patients reported similar results, with neutropenia and leukopenia rates decreased by more than 20% in the group receiving evening doses of the drug [54]. In addition to addressing the effects of chemotherapy delivered at isolated timepoints, studies have examined differences between chronomodulated variable rate versus continuous drug administration. For example, a clinical study including 54 patients compared clinical outcomes between those receiving a chronomodulated, variable rate infusion of the chemotherapeutic floxuridine (FUDR) to those receiving a continuous infusion. In this study, FUDR was delivered at varying doses for four, six-hour daily segments for a total of two weeks. The greatest dose, 68%, was delivered between 1500 h and 2100 h. The dose tolerance of FUDR was increased by 45% for patients receiving chrono-chemotherapy compared to those receiving a continuous infusion [55]. In addition, those patients also reported fewer undesirable effects such as nausea, vomiting, and diarrhea [55]. Variable dose chrono-chemotherapy reduces the adverse effects of other drugs as well, such as 5-Fluorouracil (5-FU) [42]. One study including 92 patients with colorectal cancer reported that not only was the tolerability of 5-FU increased in patients receiving the drug at the cycle peak (0400 h), but they also experienced an increased median survival rate compared to patients who received constant rate infusions [42]. These results are in line with those from other studies which suggest that the 5-FU-degrading enzyme dihydropirimide dehydrogenase (DPD) is lowest at 0400 h; thus, an optimal dose at 0400 h may coincide with the highest 5-FU efficacy [30].

As demonstrated in the aforementioned study, the efficacy of the drug and tumor response can be maximized by optimizing the schedule of chemotherapy delivery according to the circadian cycle [30]. In two separate phase III clinical trials, a morning administration of the popular chemotherapeutic agent doxorubicin was associated with an over 30% increase in five-year survival rates among ovarian cancer patients [42]. As further evidence for the timed delivery of chemotherapeutics, a study examining the effects of chrono-chemotherapy in patients with metastatic endometrial cancer found that morning administration (0600 h) of either pirarubicin or doxorubicin followed by an evening dose (1800 h) of cisplatin resulted in an objective response rate of nearly 60% [48]. More recently, a systematic review summarizing the effects of chrono-chemotherapy on drug toxicity reported that 14 out of 18 studies reported a reduction in toxicity for patients receiving chrono-chemotherapy, with one study reporting increased toxicity [56]. However, in the study reporting increased toxicity, patients in the chronomodulated group were administered 5-FU at a two times higher dose (600 mg/m^2^) than patients in the control treatment group (300 mg/m^2^) [56]. In contrast, three of the reviewed studies reported an increased objective response rate or prolonged period of treatment viability, suggesting that optimizing chemotherapy delivery may not only increase the efficacy of the drug but may also increase the length of time that a drug may be used before the patient develops drug resistance [56]. Foundational studies examining the therapeutic potential of circadian-dependent chemotherapy administration also provide additional evidence [57]. Administering cisplatin at ZT10 reduced tumor growth compared to mice that received the injection at ZT22; however, disrupted circadian rhythms (modeling “jet lag” in this study via an 8 h phase advance) abolishes the therapeutic advantage of time-dependent dosing [25]. In a model of pancreatic carcinoma, the efficacy and tolerability of the drug docetaxel were reported to be highest in the latter half of the inactive phase [58]. Remarkably, *in silico* models have also illustrated the therapeutic effect of timed chemotherapy dosing. A study examining the effect of chrono-chemotherapy in cells modeling early-stage colon cancer demonstrated that for samples in which chemotherapy was administered between 0400 h and 0800 h, cancerous cells were targeted in 100% of samples. In contrast, only 18% of the samples that were administered chemotherapy between 2000 h and 0000 h exhibited a mutant cell treatment response [59].

More recently, foundational and clinical studies have examined the effects of chrono-chemotherapy on brain metastasis and glioblastoma tumors. Chemotherapy has traditionally been considered to be less efficacious in the treatment of brain tumors, likely due to difficulty traversing the blood–brain barrier (BBB). However, studies have demonstrated that the permeability of the BBB is altered across the circadian cycle. Using this logic, a study was conducted to determine whether chronomodulated, peripherally administered chemotherapy displays increased efficacy in the treatment of brain metastases of breast cancer (BMBC) [57]. To address this question, concentrations of the chemotherapeutic agent ^14^C-paclitaxel were assessed in the brains of tumor-bearing mice at four timepoints across a 24 h period. They reported a time-of-day effect in which ^14^C-paclitaxel levels peaked during the mid-dark phase (ZT17) and were lowest at the start of the light phase (ZT0). A second study revealed this to be functionally significant, as tumor-bearing mice that received ^14^C-paclitaxel at ZT17 demonstrated a protracted period prior to exhibiting neurological deficits compared to mice that received chemotherapy at ZT0 [57]. These results were replicated in multiple aggressive metastatic cell lines, providing evidence for the use of chrono-chemotherapy for BMBC. Similar results have also been reported in clinical research. In a retrospective study of 166 patients with glioblastoma tumors, outcomes of patients that received morning infusions were compared to those of patients receiving evening infusions of the chemotherapeutic temozolomide (TMZ) [60]. Researchers also performed tissue analyses to determine the methylation status of the O-6 Methylguanine-DNA methyltransferase (MGMT), a protein that is chrono-regulated and plays a key role in repairing TMZ-induced DNA damage [60]. Whereas no differences in overall survival between patients that received morning versus evening TMZ infusions were observed alone, among patients with MGMT methylation, morning infusions increased overall survival by six months [60]. Taken together, these studies suggest that chrono-chemotherapy may serve as a promising treatment for both primary and metastatic brain tumors.

Currently, there is a very limited number of studies examining the use of nanoparticle drug delivery in chrono-chemotherapy. Nanoparticle delivery can change aspects of a drug’s kinetics in pharmacologically advantageous ways as well as increase target specificity [61,62]. In one study, the authors designed a nanoparticle vehicle to administer paclitaxel (PTX-NP) [61]. Although the study corroborated varying efficacies of paclitaxel alone throughout the day, it also demonstrated that nanoparticle delivery further increased anti-tumor efficacy (significantly more so than paclitaxel alone) at 15 h after light onset (HALO; 2200 h). Administration of PTX-NP at 2200 h resulted in an increased number of apoptotic cells and tumor growth inhibition (*in vivo*), as well as reducing the number of Ki67^+^ tumor cells (*in vivo*) and their cell viability (*in vitro*) versus paclitaxel alone [61]. These data suggest that nanoparticle drug delivery could have a significant impact on the efficacy of chrono-chemotherapy. Pending further studies, the use of nanoparticles for delivery of chemotherapeutics could potentially boost not only drug tolerability, but also the anti-tumor efficacy of chronotherapy.

Although evidence suggesting a therapeutic role for chrono-chemotherapy is increasing, several considerations must be addressed by researchers pursuing this approach. First, it is important to note that much of the currently available research on chrono-chemotherapy has focused primarily on colorectal cancers and that studies examining the effects of chrono-chemotherapy for other types of cancers are scarce. This is especially problematic when considering the evidence that the timing of DNA synthesis is modulated in part by the location of the primary tumor [48]. Given the functionally variable development and metastasis of different cancers, research must expand to include a broader representation of tumor types in order to determine whether tumors of various origin are differentially influenced by circadian-modulated chemotherapy administration. Additionally, the types of pharmaceutical agents used in these studies are limited. Common drugs such as doxorubicin, 5-FU, and cisplatin have been extensively evaluated in chrono-chemotherapy studies; other chemotherapeutic drugs should also be evaluated for time-of-day effects. Failure to examine the chronomodulation of a wide array of available pharmaceuticals may result in a missed opportunity to uncover clinically promising treatments. As an example, the chemotherapeutic oxaliplatin was discontinued following phase I of a clinical trial due to high levels of adverse effects. Interestingly, the negative effects associated with this drug were later discovered to be dependent on the time of administration [30]. Thus, chrono-chemotherapy should not only be considered for its therapeutic potential of existing drugs but should also be recognized as a primary variable when assessing the safety and efficacy of novel treatments in anti-cancer drug research. In addition, the combination of multiple drugs used to treat tumors should be considered by researchers. Given that chemotherapy is less than 10% effective when used as a stand-alone treatment for advanced cancers [44], and that many chemotherapeutics are used in tandem, chrono-chemotherapy studies should account for how these drugs are affected by circadian rhythms in conjunction to determine the most efficacious approach. Further, because some chemotherapy drugs, such as paclitaxel, alter the endogenous circadian rhythms directly [30], recognizing the chronomodulatory effects of the drug itself is an important consideration in chronotherapy research. Finally, the modality of drug administration is a factor that should be examined. Although the majority of clinical studies have utilized intravenous drug administration models, the use of oral cytotoxic medications is growing among cancer patient populations and should therefore be a high-priority target of future chrono-therapeutic research [63].

## 4. Chrono-Radiotherapy for Solid Tumor Treatment

There is a growing interest in chronomodulated radiotherapy, although the data resulting from this emerging area of research are limited and often inconsistent [42]. A further complication of efforts to elucidate the therapeutic potential of chrono-radiotherapy is the fact that many studies are performed retrospectively with differing methodological approaches. Additionally, a large number of patients included in studies examining the effect of chronotherapeutic radiotherapy were undergoing other anti-cancer treatment regimens (e.g., chemotherapy) during the trials, which potentially confounds the results. Nonetheless, it is well established that the radiosensitivity of cancer cells differs based on time of day [30]. Despite inconsistencies between study results, considered together, the evidence suggests that chronomodulating radiotherapy treatment can greatly improve patient outcomes. Thus, chrono-radiotherapy remains an understudied, yet important, topic worthy of continued research.

Similar to chrono-chemotherapy, chrono-radiotherapy offers the possibility of reduced toxicity and improved clinical outcomes in some cases. In adenocarcinoma models, administration of radiotherapy in the morning is associated with reduced toxicity [46]. Similarly, researchers have reported that for men with prostate cancer, high-dose radiotherapy (HDRT) treatment delivered prior to 1700 h resulted in increased six-year survival rates compared to those who received HDRT at night, even after controlling for disease progression, age, and hormone therapy [59]. In a different prostate cancer study, patients that received photo beam radiotherapy between 0830 h and 1030 h had fewer lower urinary tract symptoms and double the quality-of-life score compared to patients that received evening doses [59]. A recent review consisting of nine chrono-radiotherapy studies involving individuals with head and neck cancer reported that chronomodulating treatment was associated with reduced radiotherapy toxicity, despite no differences in treatment response observed across four of the studies [64]. Rates of treatment-induced oral mucositis were also analyzed, and whereas no differences were reported between groups in overall prevalence, the latency at which mucositis developed was significantly increased in the group receiving afternoon radiotherapy [65]. Finally, a study examining seasonal effects of radiotherapy in head and neck cancer patients suggested that those who received radiotherapy in the dark months had increased incidences of radiotoxicity [65].

One retrospective review examining data from 109 glioblastoma patients that received either morning or afternoon radiotherapy reported no differences in survival rates or toxicity levels among individuals [66]. Although some studies have suggested that morning radiotherapy treatment may increase survival for patients with brain metastases [42], this inconsistency highlights that origin-specific tumor cells are differentially affected by anti-cancer treatment. Thus, one possible explanation for these results is that high-grade gliomas may be more resistant to chronomodulated radiotherapy [46]. Further, this study included 62 males and 47 females. When taken in consideration with the previously demonstrated effect of sex on radiotherapy treatment response [46], it is possible that sex differences confounded the ability to observe a chronotherapeutic effect. Evidence for the therapeutic effect of chrono-radiotherapy has also varied in breast and ovarian cancer studies. For example, whereas one study examining radiotherapy effects in breast cancer patients reported that morning administration was associated with more severe adverse outcomes, another study indicated that afternoon radiotherapy resulted in elevated adverse skin effects [42,65]. Perhaps the study providing the most concrete evidence in support of the therapeutic role of chrono-radiotherapy is a retrospective review including 256 rectal cancer patients. In this study, patients who received neoadjuvant radiotherapy in the afternoon (after 1200 h) displayed an increased tumor response [67]. However, women were not as likely to achieve the same therapeutic response as men for some measures [67]. Patients displaying the greatest reduction in tumor volume started the study with smaller tumor sizes [67]. However, this effect was not studied in depth, but rather observed as a secondary outcome. Further research should examine the effect of initial tumor volume on the efficacy of chronotherapy.

In summary, the field of chronotherapeutics for radiotherapy remains largely unspecified. Studies incorporating the use of chrono-radiotherapy should focus on building prospective studies with well-designed methods aimed at increasing the rigor and replicability of studies in this field. Once consistent results are produced, existing, albeit limited, evidence suggests that sex-dependent responses to chrono-radiotherapy may provide an interesting avenue for future research to address.

## 5. Additional Chronotherapies for Solid Tumor Treatment

Traditional cancer treatments such as surgery and chemo- and radiotherapies are among the most common defenses against solid tumors. However, more modern and specific treatments are sometimes used to supplement or replace traditional therapies. There is currently a dearth of clinical trials examining the effects of the internal clock on these treatment regiments. Here we review the chronobiological effects of alternative classes of anti-tumor therapies.

### 5.1. Hormone Therapy

Hormone-modulating therapies (e.g., aromatase inhibitors, hormone receptor antagonists, etc.) are often used to treat tumors originating from endocrine tissue, such as breast, prostate, and adrenal cancers. These drugs commonly work by blocking pro-tumor hormone signaling or by rendering the hormone itself nonfunctional. In common with many other tumor treatment options, these drugs are systemic and therefore cause significant and uncomfortable side effects such as hot flashes, sexual dysfunction (e.g., decreased libido, erectile dysfunction, etc.), and gynecomastia. Because of this, patient non-compliance is remarkably high for these drugs. Indeed, a 2021 metanalysis spanning from 1947 to August 2020 found that non-compliance for anti-tumor hormone therapies ranged from 41 to 72% [31]. However, clinical trials for chronotherapeutic regimens of anti-tumor hormone therapies are severely lacking, and most chronobiological “evidence” is anecdotal [31]. The major issues with non-compliance and non-persistence in the clinic should act as an alarm for researchers. Because of the particularly devastating nature of the side effects of hormone therapy in cancer treatment, chronotherapeutic trials are desperately needed. Given the previous success of trials and studies of reducing anti-drug side effects (toxicity) with chronotherapy [29,30,42,48], it is crucial that physicians and foundational researchers work to implement this treatment strategy for patients receiving hormone therapy for cancer in the near future.

### 5.2. Tyrosine Kinase Inhibitors

Tyrosine kinase inhibitors (TKIs) are anti-cancer drugs that inhibit signaling cascades via receptor tyrosine kinases (RTKs). Potentially oncogenic RTKs are often receptors for growth factors such as vascular endothelial growth factor (VEGF), epidermal growth factor (EGF), and fibroblast growth factor (FGF). Overexpression of these receptors can lead to unrestrained progression through the cell cycle and is often associated with more aggressive and metastasis-forming cancers [68]. A 2009 phase II clinical trial examined a loosely chronomodulated regiment of the drug Sunitinib (Stutent^®^), which is commonly used to treat advanced or metastatic renal cell cancer, pancreatic neuroendocrine tumors, and treatment-resistant gastrointestinal stromal tumors [69,70]. Sunitinib specifically disrupts receptors for VEGF, platelet-derived growth factor (PDGF), stem cell factor (SCF), and colony-stimulating factor (CSF), as well as fms-like tyrosine kinase 3 (FLT-3). In the trial, 107 patients with metastatic renal cell carcinoma were administered 37.5 mg of Sunitinib (with doses adjusted based on individual tolerance throughout the study) in the morning or the evening. No statistical significance was observed in efficacy, tolerance, or quality of life between morning and evening administration [70]. However, only 22% (24/107) of participants completed the trial, and the Sunitinib dosage was not consistent throughout the study, undermining the rigor of the study for assessing chronotherapy [70]. A similar study was conducted in which patients with imatinib-resistant/intolerant gastrointestinal stromal tumors were given a consistent dose (37.5 mg/day) of Sunitinib in either the morning or evening. This study likewise indicated no significant difference in anti-tumor efficacy between morning or evening administration [71].

Contrary to clinical trial data, a study conducted in male New Zealand rabbits demonstrated a significant difference in morning versus evening administration [69]. Simple oral administration of 25 mg of Sunitinib in the animals at 0800 h or 2000 h resulted in significantly higher serum levels in animals injected at the former timepoint for the first 24 h, with peak levels being over two times higher when administered at 0800 h compared to 2000 h. The same was true for SU12662, the main metabolite of Sunitinib [69]. Given the influence of daily rhythms on metabolism, these data are to be expected. Whereas these data support a potential chronotherapeutic benefit for Sunitinib administration, parameters of end anti-tumor efficacy were not directly measured in the study. It is possible that a similar trend occurred in the serum of the aforementioned clinical trial patients [70,71], however no tangible anti-tumor benefits were observed in time-of-day administration. In-depth foundational science studies on the chronomodulation of TKI administration are needed to build a stronger basis on which clinical trials can be developed.

### 5.3. Antiangiogenic Therapy

Primary solid tumor and metastasis formation require an abundance of nutrients and growth factors to support rapid proliferation. The most efficient way to fulfill these requirements for a solid tumor is to stimulate the growth of new blood vessels, the conduit for these factors, throughout the tumor. Because this phenomenon often accompanies the generation of solid tumors, drugs inhibiting angiogenesis are used to treat some cancers. While not prescribed as often as chemo- or radiotherapy in the clinic, antiangiogenics represent a group of drugs targeting different components of nascent vasculature formation that can slow the progression of some solid tumors [72].

In one of only a couple of studies investigating a chronotherapeutic approach to antiangiogenic therapy in cancer treatment, the authors administered 30 mg/kg of TNP-470, a synthetic form of the antiangiogenic drug fumagillin, every other day at either 0700 h or 1900 h for 21 days [73]. TNP-470 works by inhibiting type II methionine aminopeptidase (MetAP-2), which in turn arrests the cell cycle in endothelial cells. The experiment was performed in three different solid tumor models (Sarcoma 180, Lewis lung carcinoma, and B16 melanoma) in male ICR mice. In all three models, administration of the drug at 0700 h resulted in significantly smaller tumor sizes for the duration of the study when compared to injection at 1900 h or the vehicle [73]. Furthermore, mice injected at 0700 h had significantly decreased MetAP-2 activity when compared to vehicle-injected mice. Interestingly, animals injected at 0700 h also had significantly higher levels of serum TNP-470 at 15 min post-administration than those receiving the drug at 1900 h [73]. These data not only demonstrate an increase in anti-tumor efficacy, but also indicate advantageous chronomodulated pharmacokinetics at 0700 h.

More recently, a 2012 study investigated the chronobiology of targeting the angiogenic factor VEGF using a unique approach [74]. The study examines the use of photodynamic therapy (PDT), a modern form of tumor treatment that utilizes light-activated drugs (photosensitizers). Advantages to this method include higher specificity, decreased invasiveness, and milder anti-drug reactions when compared to traditional treatments (i.e., chemo- and radiotherapies). Previous to this study, the authors had synthesized a novel photosensitizer in which hematoporphyrin was immunoconjugated with anti-VEGF antibodies. In the current study under discussion, this novel compound was administered in accordance with the daily oscillations in VEGF secretion by tumor cells in both Lewis lung carcinoma and sarcoma 180 murine models [74]. The VEGF content (ng/mg protein) peaked at 1400 h in both tumor models, and these data corresponded with the time of peak fluorescence following drug administration, indicating the validity of the photosensitizer. Furthermore, Lewis carcinomas developed to a significantly lower weight when the drug was administered at 1400 h when compared to drug administration at 0200 h and control administration of non-conjugated hematoporphyrin. Sarcoma 180 tumors displayed significantly decreased diameters from days 9 to 12 (end of study) when treated with the photosensitizer at 1400 h as opposed to at 0200 h or following administration of free hematoporphyrin [74]. Further research involving more parameters examining tumor growth and different variants of PDT (e.g., tissue penetration of light source) is needed to expand upon this study. In the past decade, more sophisticated techniques have been used to measure tumor growth and formation. Further experimentation could potentially elucidate tumor growth mechanisms as well as advocate for PDT chronobiological clinical trials.

### 5.4. Immunotherapy

There are currently very few studies highlighting the endogenous clock of the immune system. Although the SCN ultimately sets the clock for the whole body, different cell and tissue types have unique oscillations of clock genes such as *Bmal1*. CD80, a costimulatory molecule expressed by dendritic cells (DCs) to activate T cells, contains an e-box in its promoter region that is bound by BMAL1. In their study, Wang et al. demonstrate that tumor mass was significantly different based on time of engraftment using a murine melanoma model. Through experimentation in immunocompromised murine models, they indicated that this difference was driven by both innate and adaptive immunity. When *bmal1* was depleted in DCs or T cells, the time of engraftment differences between tumors were abrogated. More importantly, they demonstrated that anti-tumor vaccine efficacy was significantly enhanced when administered at specific times within the circadian cycle, and that this effect was dominant when compared to that seen in time-of-engraftment experiments [75]. These results suggest that there is a potential to treat immunogenic tumors using chrono-immunotherapy in humans.

Immune checkpoint inhibitors (ICIs) are used to treat several types of tumors including breast, renal, lung, and other cancers. These drugs work by inhibiting the development of immune tolerance, which is implemented in immune cells to avoid reaction to autoantigens and thus prevent autoimmune disease. Drugs targeting cytotoxic T lymphocyte antigen 4 (CTLA4) and programmed cell death 1 (PD-1) block cytotoxic T lymphocyte receptors that interact with cancer cell ligands to recognize them as autoantigens. Disrupting this interaction reduces T cell tolerance, thus promoting the elimination of cancer cells [76]. Given the data on adaptive immune cell oscillation [77,78], it is plausible that ICIs are good candidates for chronotherapeutic trials. Indeed, there is a growing body of evidence that supports circadian regulation of cancer immunology [79,80,81,82]. A 2021 longitudinal study examined outcomes of advanced melanoma patients when treated with ipilimumab (anti-CTLA4), nivolumab (anti-PD-1), pembrolizumab (anti-PD-1), or a combination of these drugs before evening. Patients who received their infusions after 1630 h less than 20% of the time lived significantly longer than those who were treated after 1630 h at least 20% of the time [80]. Furthermore, preliminary data indicate that a similar pattern may also be true for patients with non-small cell lung cancer [79]. Studies on ICIs are still very preliminary, and there are no ongoing clinical trials implementing chrono-immunotherapy in the treatment of solid tumors.

In 2020, a group of 28 leading experts in cancer, immunology, and neuroscience research published a brief commentary, officially defining the convergence as “cancer neuroscience” and creating a call to action in highlighting several gaps in the emerging field [83]. In a recent state-of-the-field update, the authors emphasized a need for research in the niche of neuro-immuno-oncology. Gaps in this subfield include research on how neuromodulating drugs can be utilized in cancer treatment and how this knowledge can be implemented to drive the immune system to eliminate cancer cells more efficiently [84]. Although chrono-immunotherapy is not directly addressed in this update, it is evident that a circadian approach to treatment needs to be considered. Given that different components of the immune system have distinct daily oscillations, this approach to tumor treatment may strengthen the efficacy of current and future immunotherapeutics for several types of cancer.

## 6. Conclusions

There is sufficient evidence to support future research (foundational science and clinical) of chronotherapeutic treatment of cancer. To date, most support comes from clinical and foundational science studies demonstrating time-of-day differences in adverse effects and efficacy of chemotherapeutics. Indeed, studies have demonstrated the beneficial effects of chrono-chemotherapy (Table 1) across diverse cancer types including colorectal, lung, breast, ovarian, bladder, and glioblastoma. Nevertheless, this has led to very little clinical application, as intravenous chemotherapy is most often given during typical staff working hours [85]. Initial data examining chrono-radiotherapy as an improved treatment option compared with conventional radiotherapy are somewhat promising. However, the results lack consistency across studies. Future examination should focus on building prospective studies with methods aimed at increasing the replicability of findings. Additional alternative chronotherapies (e.g., hormone therapy, TKIs, antiangiogenic therapy, immunotherapy) for the treatment of solid tumors are vastly understudied and represent an area of dire research need. Initial studies have demonstrated some positive evidence to support further investigation (Table 1).

There are remaining questions and challenges in the field that still need to be addressed. First, additional foundational science studies are needed to provide evidence for the beneficial effects of chronotherapeutics for the treatment of cancer. Particular focus should be on studies examining chrono-radiotherapy and alterative chronotherapeutics (e.g., hormone therapy, TKIs, antiangiogenic therapy, immunotherapy, etc.). Second, virtually all animal models examining chronotherapy for cancer treatment have used nocturnal rodent species. Given that humans are diurnal, animal models developed with diurnal species represent an area of need and may lead to an increase in clinical trials and easier translation to clinic. Next, the timing of cancer drug administration is rarely reported or stipulated in clinical trials or foundational science studies [85,86]. Thus, care must be taken by clinicians and researchers to report time-of-day information, which will allow for additional comparisons based on chronomodulation. Furthermore, variables such as sex and chronotype have yet to be considered in chronotherapeutic studies. Given the push toward personalized medicine, future studies should incorporate these variables within the experimental design. In sum, chronotherapeutics for cancer treatment are likely viable treatment strategies. However, further foundational science examination is needed to provide clinicians with clear data that will allow for implementation within the clinic.

## Figures and Tables

**Figure 1 pharmaceutics-15-02023-f001:**
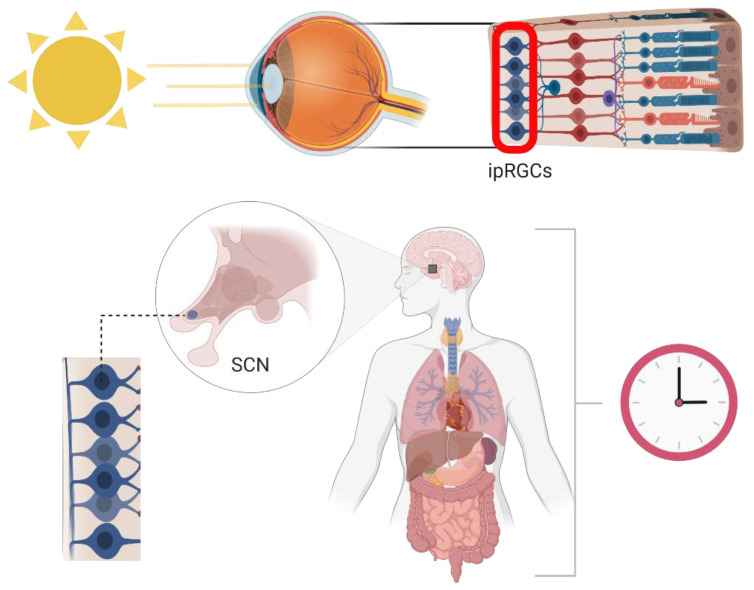
**Light is the primary zeitgeber for entrainment to a 24 h cycle in mammals.** Solar blue light (~480 nm) excites melanopsin, which in turn activates intrinsically photoreceptive retinal ganglion cells (ipRGCs) that send excitatory signals to the suprachiasmatic nucleus (SCN) of the anterior hypothalamus. Termed the central pacemaker, the SCN dictates daily oscillations in peripheral organs. Created with Biorender.com accessed on 24 July 2023.

**Figure 2 pharmaceutics-15-02023-f002:**
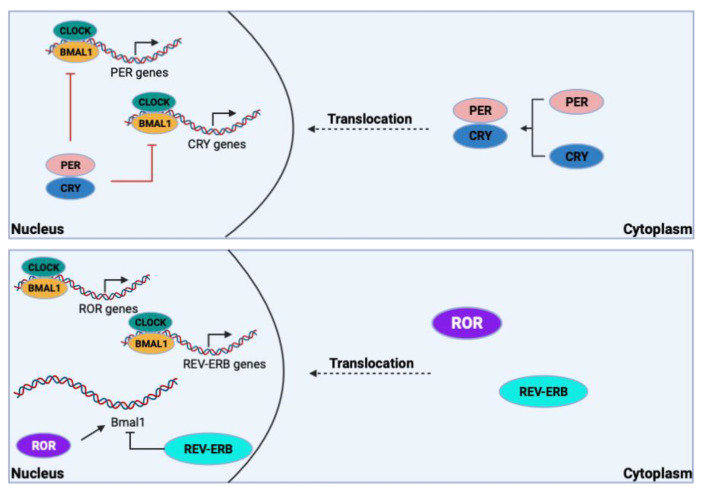
**The molecular clock is regulated by transcription–translation feedback loops.** CLOCK/BMAL1 heterodimerize, translocate to the nucleus, and regulate expression of *Per*/*Cry* genes. PER/CRY proteins accumulate in the cytoplasm where they heterodimerize and translocate to the nucleus to inhibit CLOCK/BMAL1-driven promotion of their transcription (top). The CLOCK/BMAL1 heterodimer drives expression of ROR and REV-ERB proteins. ROR and REV-ERB proteins translocate to the nucleus and either drive or inhibit BMAL1 expression, respectively (bottom). Created with Biorender.com accessed on 24 July 2023.

**Table 1 pharmaceutics-15-02023-t001:** Summary of Primary Studies Currently Reviewed.

	Tumor Type	Species	Treatment	Results
**Chrono-Chemotherapy**	Brain metastases of breast cancer [57]	Mice	13 mg/kg of paclitaxel at ZT0 or ZT17	Mice treated at ZT17 demonstrated a significant delay of neurological symptoms compared to those treated at ZT0
	Ovarian Cancer [53]	Human	60 mg/m^2^ of Adriamycin from 0600–0630 h and 60 mg/m^2^ cisplatin from 1800–1830 h (schedule A) or vise versa (schedule B)	Patients on schedule B required more dose reductions and experienced more complications (i.e., infections and bleeding)
	Various solid tumors [49]	Human	Capecitabine every day with administration of 750 mg/m^2^ at 0900 h and 1250 mg/m^2^ at 2400 h	Maximum tolerated dose was 20% higher than that of the current approved regimen (1250 mg/m^2^ bi-daily)
**Chrono-Radiotherapy**	Nasopharyngeal carcinoma [50]	Human	80 mg/m^2^ of cisplatin from 1000–2200 h; 1000 mg/m^2^ of 5-FU and 200 mg/m^2^ of citrovorum factor from 2200–1000 h for 3 days	Chronomodulation of treatment significantly reduced leukocytopenia, thrombocytopenia, and nausea/vomiting when compared to constant administration
	Nasopharyngeal carcinoma [51]	Human	DDP administration from 1000–2200 h; 5-FU administration from 2200–1000 h	Significant decrease in stomatitis during radiotherapy compared to constant administration
	Leukemia [29]	Mouse	240 mg/kg of Ara-C with higher doses earlier on that the standard regimen	Increased tolerance and survival
	Glasgow osteosarcoma [33]	Mouse	200 mg/kg of gemcitabine at HALO 11 or 23; 5 mg/kg of cisplatin 1 min or 4 h after gemcitabine	Decreased neutropenia and weight loss when gemcitabine given at HALO 11 regardless of cisplatin administration time
	Advanced non-small cell lung cancer [54]	Human	75 mg/m^2^ of docetaxel on day 1; 20 mg/m^2^ of cisplatin on days 1–4 at either 0600 h or 1800 h; 1000 mg/m^2^ of gemcitabine on days 1 and 8	Significantly reduced nausea and neutropenia with cisplatin administration at 1800 h
	Various metastases [55]	Human	Constant administration of FUDR with maximal flow rate in the late afternoon and minimal flow rate in the early morning	Patients receiving chronomodulated treatment experienced less severe gastrointestinal side effects and higher drug tolerance
	Triple negative mammary carcinoma [25]	Mouse	5 mg/kg of cisplatin at ZT10 or ZT22	Treatment at ZT22 reduced tumor growth, but this effect was negated by phase shift (jet lag)
	Glioblastoma [60]	Human	Temozolomide in the morning or evening	Patients taking Temozolomide in the morning had a higher rate of overall survival at ~5 years post-treatment
	Lung cancer [61]	Mouse	Novel nanoparticle-conjugated paclitaxel	Nanoparticle delivery resulted in significantly higher anti-tumor efficacy at HALO 15 than paclitaxel alone
	Mixed; 80% glioblastoma [66]	Human	Radiotherapy in the morning or evening	Time of day had no effect on toxicity, progression-free survival, or overall survival
	Rectal cancer [67]	Human	Radiotherapy in the morning or evening	Patients receiving treatment after 1200 h the majority of the time were significantly more likely to respond to treatment
	Naïve [69]	Rabbit	25 mg of Sunitinib at 0800 h or 2000 h	Pharmacokinetics of the drug were significantly enhanced at 2000 h
	Metastatic renal cell carcinoma [70]	Human	37.5 mg/day of sunitinib in the morning or evening	Time of day had no significant effect on efficacy, tolerance, or quality of life
	Imatinib-resistant/intolerant Gastrointestinal stromal tumor [71]	Human	37.5 mg/day of sunitinib in the morning or evening	Time of day had no significant effect on efficacy or adverse events
**Additional Chronotherapies**	Lewis lung carcinoma and sarcoma 180 [74]	Mouse	Photodynamic therapy (laser irradiation) at 0200 h or 1400 h	Tumor growth was significantly inhibited when treated during the day rather than at night
	Melanoma [75]	Mouse	Anti-tumor vaccine at ZT9 or ZT21	Tumor volume was significantly smaller following treatment in mice vaccinated at ZT9 rather than ZT21
	Non-small cell lung cancer [79]	Human	Single-agent anti-PD-1 before or after 1630 h	Patients receiving treatment after 1630 h at least 20% of the time had a significantly shorter progression-free survival than those receiving treatment after 1630 h less than 20% of the time
	Stage IV melanoma [80]	Human	Ipilimumab, nivolumab, pembrolizumab, or any combo of these before or after 1630 h	Patients receiving treatment after 1630 h at least 20% of the time had a significantly shorter overall survival than those receiving treatment after 1630 h less than 20% of the time

## Data Availability

Data sharing not applicable.

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
