# Peer review of "Chronotherapeutics for Solid Tumors"

_pharmaceutics, 2023, doi:10.3390/pharmaceutics15082023_

Round 1
Reviewer 1 Report
It is a well-structured review that provides interesting data and points to consider in future research proposal.
The molecular part concerning clocks is clear. Some reference to Zeitgeber time (ZT) would be desirable. Also, in order to improve the follow up of the article, I would suggest to include more table about the impact of chemotherapy according to the time of day when it is administered, etc. The same with other types of interventions (hormone therapy...).
Author Response
We have addressed and incorporated all of the reviewers’ and editor’s concerns in the revised manuscript. Specific responses to the reviewer and editor queries are provided on the following pages.

Reviewer 2 Report
The authors aimed to discuss the evidence for chronotherapeutic treatment for solid tumors. Specifically, studies examining chrono- chemotherapy, chrono-radiotherapy, and alternative chronotherapeutics.
Figure.1 is impressive.
The authors could elaborate a bit more, the facts regarding Cisplatin-based cancer chronotherapy directly influenced by the circadian variation of DNA repair.
Could there be rhythmic expression of clock genes after serum synchronization? 4T1-Red-F-luc cells could be synchronized by clocks in the host.
The circadian properties (differences in amplitude, phase, and period) need to be discussed in detail.
Circadian disruption promotes mammary gland associated tumor growth. Such facts need to be discussed.
The International Agency for Research on Cancer and the National Toxicology Program of the United States Department of Health and Human Services have classified night work as a potential carcinogen for several types of cancer, including breast cancer. These facts need to be discussed.
More facts need to be added regarding circadian dysregulation of DNA repair mechanisms and associated DNA damage.
Tumor volumes may also matter.
Minor grammatical errors need to be corrected.
Author Response

(The authors gave the same response as above.)

Round 2
Reviewer 2 Report
All necessary corrections were done.